# Neuroprotective Effect of Macrophage Migration Inhibitory Factor (MIF) in a Mouse Model of Ischemic Stroke

**DOI:** 10.3390/ijms23136975

**Published:** 2022-06-23

**Authors:** Ji Ae Kim, Ye Young Kim, Seung Hak Lee, Chul Jung, Mi Hee Kim, Dae Yul Kim

**Affiliations:** 1Asan Medical Center, Department of Rehabilitation Medicine, University of Ulsan College of Medicine, Seoul 05505, Korea; jirehjiae@gmail.com (J.A.K.); seunghak@gmail.com (S.H.L.); speciron90@gmail.com (C.J.); 2Asan Medical Center, Asan Institute for Life Sciences, University of Ulsan College of Medicine, Seoul 05505, Korea; dpdud5567@nate.com (Y.Y.K.); algml7843@naver.com (M.H.K.)

**Keywords:** macrophage migration inhibitory factor, stroke, in vivo, neurogenesis

## Abstract

The mechanism of the neuroprotective effect of the macrophage migration inhibitory factor (MIF) in vivo is unclear. We investigated whether the MIF promotes neurological recovery in an in vivo mouse model of ischemic stroke. Transient middle cerebral artery occlusion (MCAO) surgery was performed to make ischemic stroke mouse model. Male mice were allocated to a sham vehicle, a sham MIF, a middle cerebral artery occlusion (MCAO) vehicle, and MCAO+MIF groups. Transient MCAO (tMCAO) was performed in the MCAO groups, and the vehicle and the MIF were administered via the intracerebroventricular route. We evaluated the neurological functional scale, the rotarod test, and T2-weighted magnetic resonance imaging. The expression level of the microtubule-associated protein 2 (MAP2), Bcl2, and the brain-derived neurotrophic factor (BDNF) were further measured by Western blot assay. The Garcia test was significantly higher in the MCAO+MIF group than in the MCAO+vehicle group. The MCAO+MIF group exhibited significantly better performance on the rotarod test than the MCAO+vehicle group, which further had a significantly reduced total infarct volume on T2-weighted MRI imaging than the MCAO vehicle group. Expression levels of BDNF, and MAP2 tended to be higher in the MCAO+MIF group than in the MCAO+vehicle group. The MIF exerts a neuroprotective effect in an in vivo ischemic stroke model. The MIF facilitates neurological recovery and protects brain tissue from ischemic injury, indicating a possibility of future novel therapeutic agents for stroke patients.

## 1. Introduction

Stroke is a leading cause of mortality and disability worldwide, exerting a significant direct and indirect economic burden [1]. In 2020, the worldwide prevalence of stroke including ischemic stroke was 68.16 million [2]. Ischemic strokes, caused by insufficient blood supply to the brain, account for 80% of stroke cases. Although the prevalence of stroke is increasing, effective therapeutic agents are still lacking [3].

Ischemic stroke triggers the formation of a brain tissue necrosis core surrounded by a peripheral reversible infarction area, called a penumbra, which most neuroprotective drugs target to prevent evolution into violated tissue [4]. Brain ischemia activates inflammatory responses, apoptotic pathways, and the generation of reactive oxygen species, subsequently injuring the penumbra [5]. Therefore, novel therapeutic interventions for stroke have recently aimed to reduce the apoptotic response and manipulate inflammation [6]. The macrophage migration inhibitory factor (MIF) is a pro-inflammatory cytokine activated in response to ischemia [7]. The MIF is found in many types of cells, such as lymphocytes, neutrophils, endothelial cells, and neuronal cells. The MIF plays a variety of roles in neurological diseases, with prior research finding conflicting results [8]. The role of the MIF in ischemic stroke is controversial; some studies have suggested that the MIF has a protective function in stroke [7,9], while others have shown that the MIF aggravated stroke volume [10,11].

In previous studies, our team reported that the MIF exerted a neuroprotective effect by inducing brain-derived neurotrophic factor (BDNF) expression and reducing apoptosis [12,13]. The BDNF is a member of the neurotrophin family that promotes neuroplasticity, neurogenesis, and post-stroke motor rehabilitation [14]. In a previous study, we reported the optimal administration condition of the MIF to maximize its neuroprotective effect [15]. However, the neuroprotective effect of the MIF has not yet been investigated in in vivo models. 

The purpose of this study was to explore the neuroprotective effect of the MIF in the in vivo middle cerebral artery occlusion (MCAO) mouse model of ischemic stroke. The neuroprotection effect of the MIF in MCAO mice was investigated with behavioral testing, MRI, and Western blot.

## 2. Materials and Methods

### 2.1. Animal Models

All animal experiments were conducted with the approval of the Institutional Animal Care and Use Committee at the Asan Medical Center (2019-02-162). Experimental procedures were carried out according to the guidelines of the Revised Guide for the Care and Use of Laboratory Animals (NIH guide, 1996). We categorized the mice groups based on whether the vehicle or the MIF was applied as follows: sham+Vehicle (Sham+Veh), sham+MIF (Sham+MIF), MCAO+Vehicle (MCAO+veh), MCAO+MIF (MCAO+MIF). All 10–12 week old major mice were randomly divided into 4 groups. In this experiment, 10–12 weeks old adult male c57/BL6 mice were obtained from Koatech Inc. (Pyeongtaek, Korea). The number of each group’s mice was as follows: MCAO+veh: 14, MCAO+MIF: 14, sham+veh: 15, sham+MIF: 15. Total number of used mice was 58.

### 2.2. Procedure for MCA Occlusion

MCA occlusion surgery and MIF/vehicle injection were conducted under isoflurane anesthesia (2% induction, 1.5% maintenance). After anesthesia, the mice were placed in a supine position, and the common carotid artery (CCA) and external carotid artery (ECA) were exposed. The internal carotid artery (ICA) was exposed by incision of the neck midline. The CCA and the ECA were permanently ligated with 6/0 silk, and the ICA was temporarily clamped. Subsequently, the CCA was incised with microscissors to insert a silicon-coated filament into the middle cerebral artery (MCA), avoiding the pterygopalatine artery (PPA). The filament remained in the MCA for 60 min. After occlusion, the filament was removed carefully, and the CCA was tied for hemostasis. After closing the surgical neck incision, mice were allowed to recover under an infrared lamp [16]. The sham-operation group underwent the same surgical procedure without right middle cerebral artery occlusion [17].

### 2.3. Intracerebroventricular Injection of MIF or Vehicle by Stereotaxic Frame 

Due to the presence of the blood–brain barrier(BBB), the therapeutic agents were not delivered properly [18]. Therefore, referring to previous research, we determined an intracerebroventricular (ICV) route that can reach the brain without penetrating the BBB [7]. ICV injection of the MIF was administrated only once one day after performing MCA occlusion surgery. MCA occlusion and sham-operation mice were administered 0.9 ng/1 uL MIF (CSB-AP000501MO, Cusabio Biotech Co. Ltd., Wuhan, China) by intracerebroventricular (ICV) injection using a stereotaxic frame (JD-SI-02, JEUNGDO bio & Plant Co., Ltd., Seoul, Korea) and a syringe pump (JD-SI-03, Harvard Apparatus, Natick, MA, USA) comprising a 26-gauge stainless-steel needle 3 inches in length (701N; Hamilton Bonaduz AG, Switzerland). This concentration was determined considering two factors: (1) considering CSF circulation, we used twice the concentration of 60 ng/mL used in the previous in vitro study [15]; (2) the CSF volume in male c57/BL6 mice was calculated in a prior study [19]. For setting 120 ng/mL final concentration of ventricle, our researcher added 6.5 uL of all ventricles volume and 1 uL of MIF injection volume. Then, the value was multiplied by 120 ng/mL. Through this calculation, the amount of MIF to be administered to one mouse was 0.9 ng, and the volume was determined to be 1 uL. In addition, mice in the vehicle group were injected with a phosphate buffer solution (PBS) containing 1% Bovine serum albumin (BSA). After being anesthetized with isoflurane, the mice were placed in a stereotaxic frame, and an incision was made in the midline, equidistant from each eye. The stereotaxic coordinates were 0.3 mm posterior to the bregma; 1.0 mm right of the midline; 3.0 mm depth. The MIF or the vehicle was delivered intracerebroventricularly.

### 2.4. Neurological and Rotarod Test

The neurobehavioral effect of cerebral ischemia was evaluated after MCA occlusion on postoperative day (POD) 0, POD 1, POD 3, and POD 7. The baseline function was evaluated as POD 0. The Garcia test and the rotarod test were used for evaluation.

### 2.5. Neurological Test

The Garcia score is a neurological test to verify the occurrence of cerebral ischemia in mice. We used an 18-point score adapted from that developed for stroke by Garcia et al. [20]. The maximum score of 18 points is determined by examining spontaneous activity, symmetry in the movement of four limbs, forepaw outstretching, climbing, body proprioception, and vibrissae touch.

### 2.6. Rotarod Test

The rotarod test was conducted to evaluate each group’s motor and balance function. The rotor used for the test rotates force motor activity, speeding up every 30 s from 2.5 to 45 rpm. The riding time (seconds) was measured for up to 600 s. The longer the time held on the rotor, the better the motor function and balance function of the mice. The experiment was repeated in triplicate, with the longest time taken as the final score [21]. 

### 2.7. MRI

An MRI was conducted under 1% isoflurane anesthesia. All images were obtained using a 9.4 T/160 mm animal MRI system (Agilent Technologies, Santa Clara, CA, USA). Radiofrequency excitation and signal detection were performed using a 72 mm right-angled (quadrature) volume coil and two-channel phase arrangement coils, respectively. The pieces of the axis corresponding to the coronal image of the neuroanatomic axis were collected from the cervical spinal cord to the olfactory bulb. MRI imaging parameters (T2-weighted images [T2WI], and diffusion weighted imaging [DWI], fractional anisotropy [FA], and the apparent diffusion coefficient [ADC]) were obtained 7 days after MCA occlusion (POD7). The protocol included a T2WI (TR = 4000 ms; TE = 38.40 ms; ESP = 12.80 ms, slice thickness = 0.80 mm; matrix = 256 × 256 [no gap]), a DWI (TR 2000 ms; TE = 23.11 ms; slice thickness = 0.80 mm), and a diffusion tensor image acquired by Jones30_b1000 (TR = 3750.00 ms; TE = 47.57 ms; slice thickness = 0.80 mm; matrix = 96 × 96)].

### 2.8. Infarction Volume

The infarction volume was measured using an Image J program with the following formula: correct infarct volume (mm^3^) = [contralateral side hemisphere volume—(lesion side hemisphere volume—ischemic lesion volume)]/contralateral side hemisphere volume × 10,019 [22].

### 2.9. Western Blot

On the postoperative day, the mice were sacrificed in a carbon dioxide chamber after undergoing an MRI. The brains were then harvested and frozen in liquid nitrogen. The brains were washed twice for 3 min in PBS. Subsequently, 200–300 uL protein extract solution (RIPA, Thermo Fisher Scientific, Waltham, MA, USA) with protease inhibitor was added to the brain tissue and reacted at 4 °C for 30 min. The brains were then crushed with a disposable homogenizer and centrifuged at 4 °C, 16,000 rcf for 15 min to separate the supernatant. Protein in the supernatant was measured using the Bradford method, and each sample was normalized to a concentration of 1 μg/μL. Subsequently, sodium dodecyl sulfate-polyacrylamide gel electrophoresis (SDS-PAGE) was performed. Blots were run using a Bio-Rad system (Bio-Rad, Hercules, CA, USA) and transferred to a nitrocellulose membrane for 13 min at 2.5 A and 25 V. After the transfer process, membranes were blocked for 1 h in 5% skim milk in TBS-T and then incubated with the primary antibody overnight at 4 °C. The primary antibodies used included: Bcl-2 (ab7973, Abcam, Cambridge, UK, RRID:AB_306187, 1:1000 diluted in TBS-T), mature BDNF (ab108319, Abcam, RRID:AB_10862052, 1:1000 diluted), MAP2 (MA5-12826, Thermo Fisher Scientific, Waltham, MA, USA, RRID:AB_10976831, 1:1000 diluted), caspase-3 (ab4051, Abcam, RRID:AB_304243, 1:1000 diluted) and Bax ab53154, Abcam, RRID:AB_867795, 1:1000 diluted), IL-6(ab9324, Abcam, RRID:AB_307175, 1:1000 diluted), IL-1β (ab234437, Abcam, 1:1000 diluted), TNF-α (ab183218, Abcam, RRID:AB_2889388, 1:1000 diluted). Then, they were washed twice for 10 min with TBS-T, and the HRP-conjugated secondary antibody was attached for 1 h. MAP2, mature BDNF, IL-6 reacted with mouse IgG antibody (GTX213111-01, GeneTax, Irvine, CA, USA,1:5000 diluted), and Bax, IL-1β, TNF-α reacted with goat anti-rabbit IgG (GTX213110-01, Enzo Life Science, New York, NY, USA). After the secondary antibody reaction, the membranes were washed three times for 10 min in TBS-T. Membranes were visualized with Detech SuperSignal™ West Femto Maximum Sensitivity Substrate (34095, Thermo Scientific, Waltham, MA, USA) under the ImageQuent LAS 4000 (GE Healthcare Life Sciences, Logan, UT, USA) imager. 

### 2.10. Statistical Analysis

All statistical analyses were performed using SPSS for Windows version 25 (IBM SPSS Inc., Chicago, IL, USA). Differences were considered statistically significant at *p* < 0.05. Data are presented as the mean ± standard deviation (SD). One-way analysis of variance (ANOVA) and Tukey’s post hoc test were used to analyze between-group differences for the Garcia score and the rotarod. For infarction volume, comparison *t*-test was used for MCAO groups comparison. For Western blot analysis, a *t*-test was used to compare the expression level of each marker in the sham group and the MCAO group, respectively. For Western blot, at least 3 samples from each group were selected, and we repeated the Western blot experiment at least 5 times. For Western group analysis, 12 independent experiments were included. 

## 3. Results

### 3.1. Behavioural Testing

#### 3.1.1. Garcia Score

The Garcia score decreased rapidly in the MCAO groups compared to the sham groups after MCA occlusion surgery (Figure 1). However, at POD3 and 7, the MCAO+MIF group’s neurological scale recovered, and the neurological scale of the. MCAO+MIF group was significantly higher than that of the MCAO+veh group (* *p* < 0.0001).

#### 3.1.2. Rotarod Test

Figure 2 demonstrates the changes in the motor function of mice with the rotarod test. Both the Sham+veh and Sham+MIF groups showed no significant impairment in their motor function. Their latency riding on the drum exceeded 500 s. Conversely, in the MCAO groups, the mean latency decreased after middle cerebral artery occlusion surgery. The riding time had somewhat recovered on post-operative day 3(POD3) in the MCAO+veh and MCAO+MIF groups. However, rotarod latency in the MCAO+veh group deteriorated on POD7, while that for the MCAO+MIF group was maintained. The latency of MCAO+MIF was significantly higher than that of the MCAO+veh group (* *p* < 0.0001). This indicates that the administration of the MIF prevents deterioration of motor function.

#### 3.1.3. MRI

An MRI was performed on POD 7. Figure 3A shows the T2-weighted image and ADC map in each group, indicating acute right hemisphere MCA territory infarction in the mice that underwent MCAO intervention. T2-weighted imaging of the MCAO+veh group showed a much higher signal intensity lesion along the right middle cerebral artery territory compared to the MCAO+MIF. When quantifying these infarction areas, the infarction volume was found to be significantly smaller in the MCAO+MIF group than that in the MCAO+veh group (* *p* < 0.0001).

### 3.2. Western Blot

#### 3.2.1. Neuronal Markers

Comparison of BDNF expression in the sham+MIF and sham+veh group showed no significant difference (*p* = 0.503). In the sham groups, MIF administration did not significantly alter BDNF expression. Comparison of the MCAO groups showed that the expression of BDNF increased significantly in the MCAO+MIF group compared to that in the MCAO+veh group (* *p* = 0.042) (Figure 4A,B). This indicates that administration of the MIF increases BDNF expression following ischemic insult. The expression of MAP2, a neuronal marker, was evaluated to verify recovery and the viability of neurons. The expression of MAP2 was higher in the MCAO+MIF group than in the MCAO+veh group (* *p* < 0.0001), indicating that neuronal viability may be increased if the MIF were administered even after experiencing ischemia (Figure 4C,D).

#### 3.2.2. Apoptosis Markers

When the MIF was administered to the MCAO model mice, the expression of caspase-3 decreased; however, this was not significant (*p* = 0.082) (Figure 5A,B). Western blot further showed that Bax was not fully expressed in the sham groups but was strongly expressed under ischemia in the MCAO+veh group. Administration of the MIF to the MCAO mice reduced the expression of Bax non-significantly (*p* = 0.082) (Figure 5C,D). No significant difference in Bcl2 expression was observed in the sham groups (*p* = 0.514). However, MIF administration increased the expression of Bcl2 in the MCAO mice, although the difference was not significant (*p* = 0.107) (Figure 5E,F). The Bax:Bcl2 ratio was significantly higher in the MCAO+MIF group than in the MCAO+veh group (* *p* < 0.0001), indicating the neuronal protective effect of MIF administration after ischemia (Figure 5G).

#### 3.2.3. Pro-Inflammatory Markers

The expression level of IL-1β was significantly higher in the MCAO+MIF group than in the MCAO+veh group (* *p* < 0.003) (Figure 6 A,B). Furthermore, the expression level of IL-6 was significantly higher than in the MCAO+veh group (* *p* < 0.004) (Figure 6 E,F). In the case of TNF-α expression level, there was no significant difference between the two groups (*p* = 0.383) (Figure 6 C,D).

## 4. Discussion

In this study, we found that the infarction volume, neurological scale, and motor function of MCAO-model mice improved with MIF administration. Western blotting further revealed that the expression of MAP2, Bcl2, BDNF, and IL-6 increased, while the Bax/Bcl2 ratio decreased in the ischemic area of MCAO mice with MIF administration.

Previous studies have shown that exercise-induced MIF may aid in motor and neurological recovery [13,23]. Chang et al. reported that neurologic scale and motor strength improved in the early treadmill exercise MCAO group and that higher levels of MIF and BDNF were associated with better motor and neurology recovery in MCAO mice [13]. Li et al. further found that increases in the MIF lead to a lower neurologic severity score in MCAO rats, indicating that the MIF promoted neurologic function recovery. In accordance with these results, the present study revealed that MCAO mice with intracerebroventricular MIF injection demonstrated better motor recovery and neurologic function.

In addition, the infarction volume was smaller in MCAO mice treated with an MIF intracerebroventricular injection compared to the MCAO controls. Zhang et al. previously found that infarction volume was increased in the MIF knockout mice compared to that in the wild type, in an NFκB-dependent manner [24]. However, our study result contradicts a prior study that found that the MIF disrupts the blood–brain barrier, increasing permeability and increasing the infarction size. In this study, the MIF was administered intravenously, and the concentration was quite high: 3.3 µg/kg [25]. When the MIF was administered at a 10-fold lower concentration, there was no significant increase in infarction volume [25]. This result may therefore have indicated a toxic effect of MIF overdose. In the current study, we administered 120 ng/mL MIF via the ventricle, as in a previous study [15], and observed a protective effect.

Cerebral infarction results in an accumulation of inflammatory cells in the damaged tissue, with concurrent activation of the apoptosis pathway [6,26,27]. The MIF inhibits apoptosis by reducing caspase-3 activation and binding to the CDCR4 receptor [7,24]. Bcl-2 is a pro-survival protein which acts by blocking the pro-apoptotic protein Bax in the midstream of the apoptosis pathway [28,29,30]. A lower Bax/Bcl-2 ratio favors the anti-apoptotic Bcl-2 protein, thus indicating a neuroprotective function [29]. Furthermore, the Bax/Bcl-2 ratio was significantly lower in MCAO with the MIF mice, indicating that the MIF may exert its neuroprotective effect by reducing apoptosis.

The promotion of BDNF expression further indicates that the MIF exerts a neuroprotective function under ischemic conditions [12,13,15]. In a prior study, Jung et el. found that expression of BDNF increased in mice administered the MIF during oxygen and glucose deprivation/reperfusion compared to that in controls [15]. BDNF promotes neuronal survival, thus increasing synaptic growth and facilitating synaptic repair [31,32]. The BDNF expression in MCAO mice treated with the MIF was significantly higher than that in the vehicle group. BDNF may facilitate synaptic plasticity, resulting in better neurologic performance.

As a cytokine, the MIF engages in and modulates immunological reactions [33,34]. The MIF binds to CD74, a type 2 transmembrane protein, activates the MAP kinase pathway, and induces monocyte/macrophage activation [35,36,37]. Benedek et al. reported that treatment of LPS-stimulated mice with a recombinant T-cell receptor ligand could reduce the production of the inflammatory cytokines IFN-γ, IL-1α, IL-1β, and IL-6 [36]. A previous report suggested that elevated MIF and CD74 expression is linked to stroke clinical severity and causes a larger infarct size by activating and recruiting T cells [34]. However, Odysseus et al. suggested that the mechanism of immune cell recruitment by the MIF after cerebral infarction is unclear [7]. Moreover, in the MIF knockout mice, the expression level of pro-inflammatory markers and CD74+ cells was not affected by the absence of the MIF after cerebral infarction [38]. In our study, the expression levels of IL-1β and IL-6 were significantly higher in the MCAO with the MIF injection group. IL-1β and IL-6 are known to promote neurogenesis after ischemic brain damage [39], which may explain the better neurologic outcome, and smaller infarction in the MIF-injected MCAO group. Moreover, the inflammatory markers might contribute to ischemic tolerance which explains better neurologic outcome in the MCAO MIF group [40].

This study has several limitations. First, only young male mice were used; this may not reflect the real clinical setting, as stroke is commonly related with co-morbidities. Second, the sample size was relatively small, and further studies with a larger sample size are warranted. Third, the detailed mechanism of action of the MIF on apoptosis and inflammation remains unclear. Further study focusing on the mechanism of the anti-apoptotic and pro-inflammatory processes of the MIF involving activating T-cells, macrophage, and monocytes is warranted. Fourth, the MCAO model cannot explain the lacunar infarcts of which the pathophysiology prognosis is different from other ischemic cerebral infarcts; thus reproducing the animal model is difficult [41]. Further study investigating the effect of the MIF in lacunar infarction is needed. Fifth, for quantifying the expression level of the apoptosis marker and the inflammatory marker, ELISA could be a more delicate method, Further study with a more delicate quantifying method is warranted in the future. Sixth, MIF injection was performed intracerebroventricularly, and this method is quite limited in humans., Further study is needed for delivering the agent in a clinical setting. Seventh, we used the suggested optimal MIF dose from an in vitro study which was conducted on a human nueroblastoma cell. Due to different physiology between humans and mice, it might not be the most appropriate dose. However, there were significant improvements in behavioral, neurological, and imaging. Finally, this study did not evaluate the long-term effect of the MIF effects, and further study evaluating long-term effects to elucidate pro-inflammatory processes after ischemic stroke are needed.

## 5. Conclusions

This study demonstrated that the MIF could exert a neuroprotective function after ischemic cerebral infarction. MCAO model mice administered the MIF showed better outcomes in neurological and motor function tests, as well as smaller ischemic volume. This suggests that the MIF could protect neuronal cells from apoptosis while inducing the inflammatory process, facilitating neurogenesis.

## Figures and Tables

**Figure 1 ijms-23-06975-f001:**
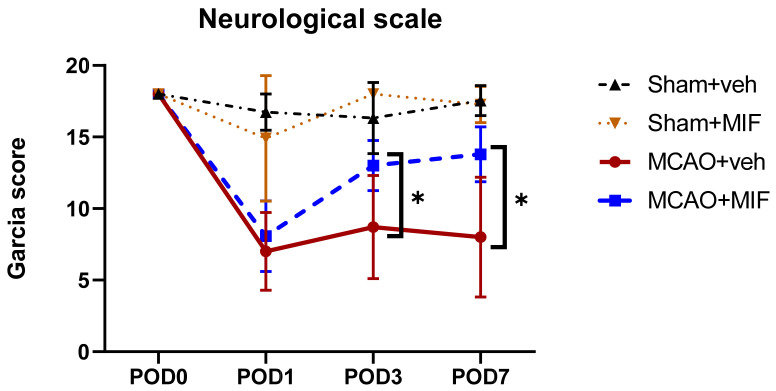
Results of the Garcia score in the MCAO+veh (*n* = 14), MCAO+MIF (*n* = 14), Sham+veh (*n* = 15), and Sham+MIF (*n* = 15) groups. At POD 1 after MCA occlusion surgery, the neurological scale of the MCAO groups decreased. At POD 3 and 7, the neurological scale of the MCAO+MIF group had recovered and was higher compared to that of the MCAO+veh group (* *p* < 0.0001). The Mean±SD values are presented here. The data were analyzed using one-way ANOVA, and post-analysis was performed using Tukey’s post hoc test. MCAO: middle cerebral artery occlusion; MIF: macrophage migration inhibitory factor.

**Figure 2 ijms-23-06975-f002:**
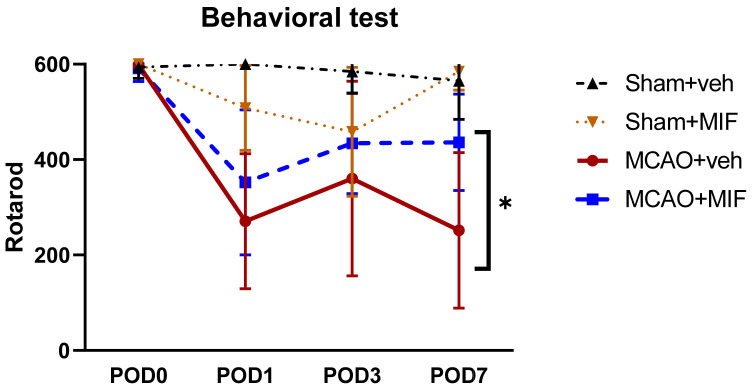
Rotarod latency. The Sham+veh (*n* = 15) and Sham+MIF (*n* = 15) groups both obtained scores of more than 500 in the rotarod test through all time points (POD0, 1, 3, 7). However, the score of mice that underwent MCA occlusion surgery decreased by POD 1. In the MCAO+veh group (*n* = 14), the latency temporarily increased slightly at POD3 but decreased again at POD7. As in the MCAO+veh group, the MCAO+MIF group (*n* = 14) exhibited a sharp decrease at POD1, but an increase occurred after MIF administration. The latency of MCAO+MIF was significantly higher compared to that in the MCAO+veh group (* *p* < 0.0001). Mean±SD values are presented. The data were analyzed using one-way ANOVA, and post-analysis was performed using Tukey’s post hoc test. MCAO: middle cerebral artery occlusion; MIF: macrophage migration inhibitory factor.

**Figure 3 ijms-23-06975-f003:**
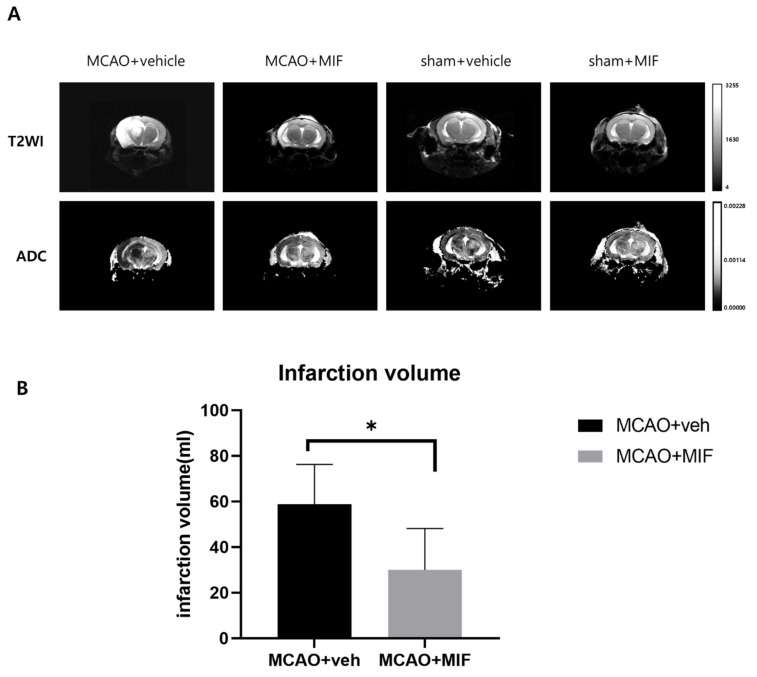
Infarction volume changes: (**A**) shows that representative images show the actual size of the infarct through T2WI, ADC; (**B**) shows that the infarction volume in the MCAO+MIF group (*n* = 14) was significantly smaller than that in the MCAO+veh group (*n* = 13), * *p* < 0.0001; the data were analyzed using a *t*-test to compare infarction size between two MCAO groups. Mean ± SD values are presented. MCAO: middle cerebral artery occlusion; MIF: macrophage migration inhibitory factor.

**Figure 4 ijms-23-06975-f004:**
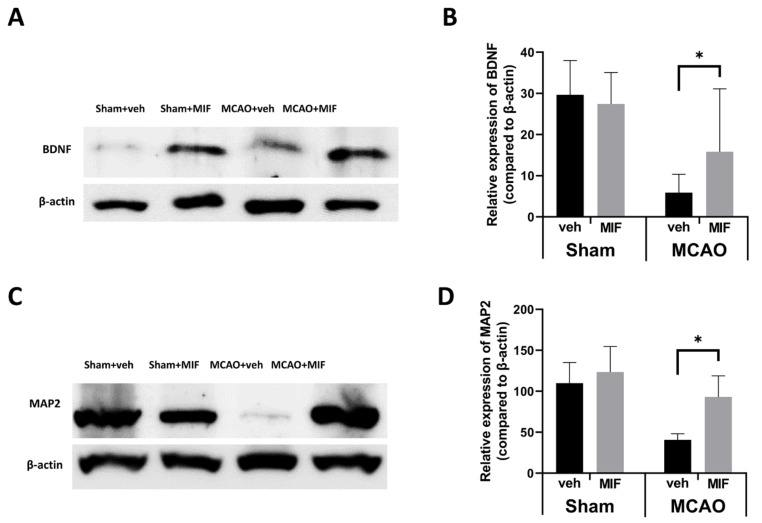
Changes in neuronal markers: (**A**,**C**) representative Western blot bands for each group are presented; (**B**) graphs showing quantification of BDNF expression in the sham and MCAO groups compared, respectively, in each group with an independent *t*-test. There was no significant change of BDNF expression after MIF injection in the sham group (*p* = 0.503); the expression of BDNF in the MCAO+MIF group was significantly higher than in the MCAO+veh group (* *p* < 0.042); (**D**) the MAP2 expression of the MCAO+MIF group was significantly increased compared to that of the MCAO+MIF group (* *p* < 0.0001); the expression level of BDNF and MAP2 in the sham and MCAO groups were analyzed using a *t*-test, respectively. Mean±SD values are presented from 12 independent experiments. MCAO: middle cerebral artery occlusion; MIF: macrophage migration inhibitory factor.

**Figure 5 ijms-23-06975-f005:**
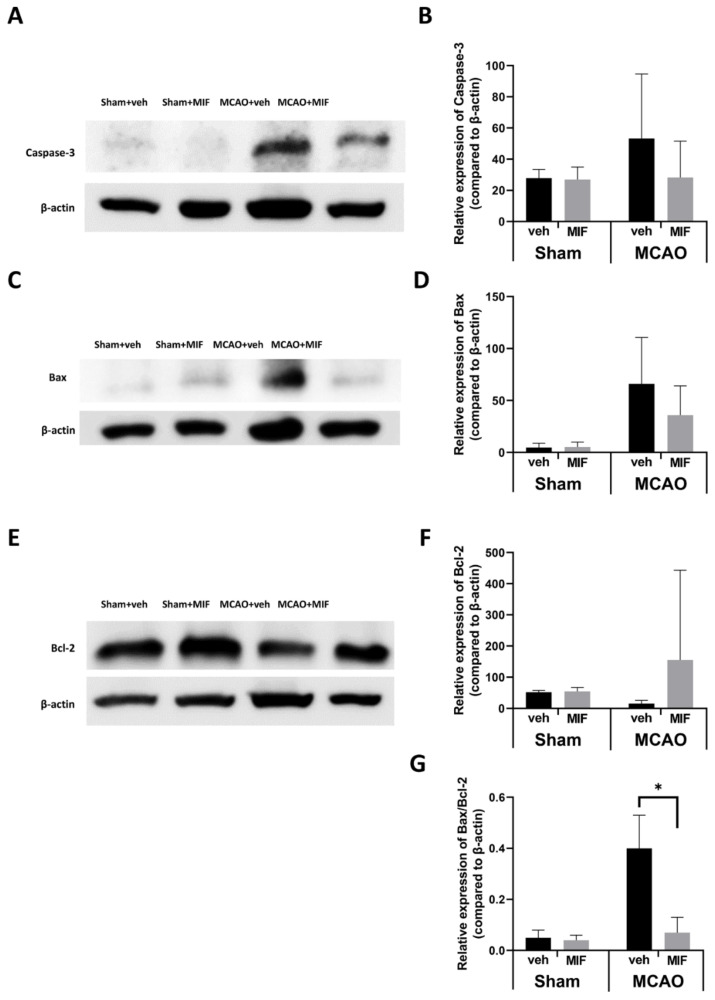
Changes in apoptosis markers: (**A**,**C**,**E**) representative Western blot bands in each group; (**B**) caspase-3 expression in the sham and MCAO groups were analyzed respectively by a *t*-test; the expression level of caspase-3 in the sham group did not change significantly with an MIF injection (*p* = 0.756); the expression level of caspase-3 in the MCAO+MIF decreased insignificantly compared to the MCAO+veh (*p* = 0.082); (**D**) The expression level of Bax decreased in the MCAO+MIF group compared to that in the MCAO+veh group; however, it was not significant (*p* = 0.082); (**F**) the expression level of Bcl2 in the sham group did not change significantly with an MIF injection (*p* = 0.514); Bcl2 expression increased in the MCAO+MIF group compared to that in the MCAO+veh group, but it was not significant (*p* = 0.107); (**G**) the Bax/Bcl-2 ratio was significantly decreased in the MCAO+MIF group compared to the MCAO+veh (* *p* < 0.0001); caspase-3, Bax, Bcl-2, and Bax/Bcl2 expression of the sham and the MCAO groups were compared, respectively, in each group using a *t*-test. Mean±SD values are presented from 12 independent experiments. MCAO: middle cerebral artery occlusion; MIF: macrophage migration inhibitory factor.

**Figure 6 ijms-23-06975-f006:**
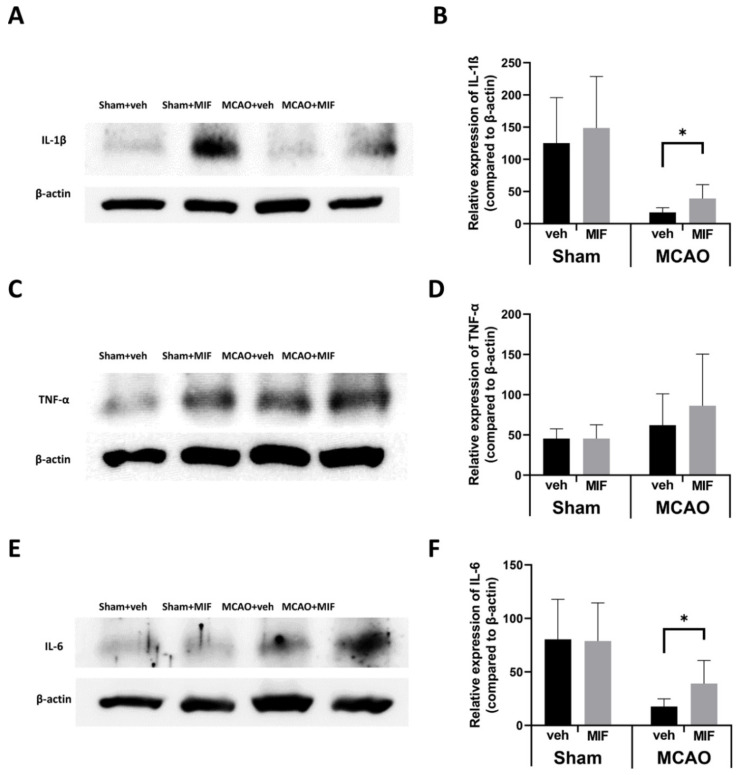
Changes of pro-inflammatory markers: (**A**,**C**,**E**) representative Western blot bands for each group; (**B**) expression level of IL-1β between the sham+MIF and the sham+veh groups was not significantly different (*p* = 0.909); the IL-1β expression in the MCAO+MIF group was significantly facilitated compared to that in the MCAO+veh group (* *p* < 0.003); (**D**) the expression level of TNF- α expression was not significantly increased following MIF administration in the MCAO group (*p* = 0.383) and in the sham group (*p* = 0.992), respectively; (**F**) IL-6 expression in the MCAO+MIF group was significantly increased compared to the MCAO+veh group (* *p* < 0.004); there was no significant difference between sham groups (*p* = 0.916). Pro-inflammatory marker expression of all sham and MACO groups was analyzed, respectively, with a *t*-test. Mean±SD values are presented from 12 independent experiments. MCAO: middle cerebral artery occlusion; MIF: macrophage migration inhibitory factor.

## Data Availability

The data are available from the corresponding author on request. For data privacy regulations, the data are not publicly available.

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
