# Peer review of "Neuroprotective Effect of Macrophage Migration Inhibitory Factor (MIF) in a Mouse Model of Ischemic Stroke"

_ijms, 2022, doi:10.3390/ijms23136975_

Round 1
Reviewer 1 Report
The manuscript entitled “Neuroprotective effect of macrophage migration inhibitory factor (MIF) in a mouse model of ischemic stroke” addresses the beneficial effects of the macrophage migration inhibitory factor, a pro-inflammatory cytokine instigated by ischemia against ischemic stroke triggered behavioral deficits and associated mechanisms. Interestingly, MIF limited the infarct size and improved behavioral performance assessed by the neurological functional scale and rotarod tests. At the molecular level, MIF augmented the prosurvival signal BDNF and the neuronal viability signal MAP2 besides lowering the Bax/Bcl2 ratio. However, it failed to significantly modify caspase-3, Bax, and Bcl2 ratio apoptotic markers. Moreover, MIF further increased the brain levels of IL-1 beta and IL-6.
Comments:
1) In Figure 5, the western blots for caspase 3 and Bax are not clear. Please, replace them with better representative blots.
Authors are also advised to use another technique for the quantification of the 3 apoptotic markers (caspase 3, Bax, and Bcl2) such as using ELISA, for example since they were not changed despite the increase in the pro-survival signal BDNF and neuronal viability signal MAP2. This point needs to be carefully addressed by the authors.
2) In Figure 6, the western blots for IL-1 beta, TNF-alpha, and IL-6 are not clear. Please, replace them with better representative blots.
Authors are also advised to use another technique for the quantification of apoptotic markers if Western is not clear. This point needs to be carefully addressed by the authors.
3) In line 258, the authors state that “but it was not significant (p = .017)”. As the p-value is less than 0.050, it should be significant. So, please, revise and correct this point in figure 5F. Also, please revise this issue since figure 5F shows a high SD which is inconsistent with the p <.017.
4) in lines 307-309, the authors state that “Our western blot showed that the expression of Bcl-2 was significantly higher in MCAO mice administered MIF intracerebroventricularly compared to that in the vehicle group”.
- The statement is not accurate and is contradictory to what the authors presented in figure 5F and in section 3.2.2. (apoptosis markers) as described by the authors themselves in lines 244 and 245, “However, MIF administration increased the expression of Bcl2 in the MCAO mice, alt- 244hough the difference was not significant (p = .107) (Figure 5E and 5F)”.
Hence, the authors are advised to remove the statement in lines 307-308. Notably, that is why the reviewer has asked the authors to try another technique such as ELISA for example in order to determine the levels of Bax and Bcl2 in brain tissues.
- Likewise, authors are advised to modify the statement in lines 24-25 accordingly by removing Bcl2 in “Expression levels of BDNF, Bcl2, and MAP2 tended to be higher in the MCAO MIF group than in the MCAO vehicle group”.
5) In section 2.1, authors are advised to specify animal species, age, gender ..etc.
- What was the total number of used animals?
- In line 70, what do the authors specifically mean by “All 10-12 major mice were randomly divided into 4 groups”.
- How many times did animals receive MIF? Is it once or more? Specify which days was MIF administered.
6) How did the authors decide about the ICV route? How is it relevant to the human administration route? In fact, humans are more common to administer therapeutic agents using the intravenous route. Authors are advised to address this point in the Material and methods section.
7) Figures 1 and 2 are confusing to readers. Authors are highly advised to use colored versions of figures 1 and 2 with distinct/different colors for each experimental group.
- A colored version of Fig. 3A is also advised.
8) Authors are advised to add statistical significance symbols to figures 1 and 2. Without statistical significance, the data are not reliable.
9) In line 167, what do the authors specifically mean by “Each experiment was repeated at least five times”? For example, in 1, for each n=14-15, this is the number of animals that give the current data, so, what is the 5-time repeat?
10) How is the dose of MIF is relevant to the human dose using the Human effective dose (HED) formula= animal dose x animal Km/ human Km (Nair AB, Jacob S. A simple practice guide for dose conversion between animals and humans. J Basic Clin Pharm. 2016 Mar;7(2):27-31). Please also provide proper citations for selecting such a dose.
11) The authors stated that “Each experiment was repeated at least five times”. The loading control was repeated in figures 4C and 5E. This is inconsistent. Please, provide different loading control.
12) In figures 4-6, how many times were the replicates for western blotting? It should be at least 3 independent experiments. However, the authors state in the legend of these figures, that n=12. Please, revise carefully. Also, after revision, correct this issue in these legends.
13) In figures 4-6, authors are advised to quantify the target proteins in the 4 experimental groups not only the last 2 experimental groups.
14) In lines 253-254, the authors state that “caspase-3 expression in the MCAO+veh (n = 12) and MCAO+MIF (n = 12) groups was analyzed by t-test. This is inappropriate since ANOVA was used herein, Tukey post hoc test should be used. Please, address this issue.
Minor comments:
15) In the abstract section (line 21), write the full name of Gar
16) In the abstract section, (line 22), the authors are advised to start the description of results in terms of the MCAO MIF group. So, the sentence should start as “The MCAO MIF group exhibited significantly better performance on the performance on the rotarod test than the 22MCAO MIF group….”
17) In section 2.3 (Intracerebroventricular injection of MIF or vehicle by stereotaxic frame). The authors are advised to clarify to readers how the administered dose for mice (0.9 ng/ul) is twice the used concentration in the in-vitro experiment (60 ng/mL=0.06 ng/ul). Please clarify this point in section 2.3. If necessary, state “almost double the used concentration”.
18) Before proceeding with one-way ANOVA, were all data checked for normality and homogeneity?
19) In line 214, authors state that “* p < .000”. Please, correct to “* p < .0001”. Please, correct this issue in the remaining figure legends.
20) In order to make all figure legends stand-alone, authors are advised to add at the end of each legend the full name of the used abbreviations.
- Also correct p = .017 to p <0.017.
21) In line 273, please correct - Also correct p = .003 to p <0.003.
22) More recent citations are advised.
Author Response
Heartily thank you heart for your all valuable comments, this manuscript improved a lot thanks to your comments. Please see the attachment for answers.

Reviewer 2 Report
The purpose of this experimental study was to explore the neuroprotective effect of Macrophage migration inhibitory factor (MIF) in the in vivo middle cerebral artery occlusion (MCAO) mouse model of ischemic stroke. The neuroprotection effect of MIF in MCAO mice was investigated with behavioral testing, MRI, and western blot. In their study, the authors found that the infarction volume, neurological scale, and motor function of MCAO-model mice improved with MIF administration, demonstrating that MIF could exert a neuroprotective function after ischemic cerebral infarction. This suggests that MIF could protect neuronal cells from apoptosis, while inducing the inflammatory process, facilitating neurogenesis. The study is potentially interesting, but can be improved if the following considerations are addressed:
1. To highlight the epidemiological importance of cerebrovascular diseases, it would be useful to mention in the Introduction the results of an epidemiologic study in Catalonia (Spain) on acute stroke (Rev Esp Cardiol 2007; 60; 573-580). In this study, the cumulative incidence of cerebrovascular diseases per 100,000 population was 218 (95% CI, 214-221) in men and 127 (95% CI, 125-128) in women.
2. Describe the acronyms used (e.g., MCAO, in the Introduction).
3. It would be helpful to mention that the previous presence of a TIA is associated -in humans- with a good early outcome in non-lacunar ischemic strokes, thus suggesting a neuroprotective effect of TIA possibly by inducing a phenomenon of ischemic tolerance (see and add this reference Cerebrovasc Dis 2004; 18. 304-311). Did the authors consider this in their experimental study?
4. The Discussion should clearly state that the pathophysiology, prognosis and clinical features of small vessel ischemic strokes are different from other acute cerebral infarcts (see and add this recent reference: Int J Mol Sci 2022; 23, 1497).
Author Response
Thanks for your bright insight about ischemic tolerance and small vessel disease. I hope you like the revised version of our manuscript.

Reviewer 3 Report
The research manuscript titled as ‘Neuroprotective effect of macrophage migration inhibitory factor (MIF)
in a mouse model of ischemic stroke’ by Kim et al. studied the neuroprotective effect of MIF in the in vivo MCAO mouse model of ischemic stroke, through behavioral testing, MRI, and western blot. They found that the infarction volume, neurological scale, and motor function of MCAO-model mice improved with MIF administration. They also found the expression of MAP2, Bcl2, BDNF, and IL-6 increased, while the BAX/Bcl2 ratio decreased in the ischemic area of MCAO mice with MIF administration. This work should be of wide interests to most researchers on neuroscience and molecular medicine etc.
This manuscript has a good logic and structure, however, there are areas that the authors may need to improve, for examples,
1. It is not known what the content of the “vehicle” is.
2. In line 85, “MCA occlusion and sham-operation mice were administered 0.9 ng/1 uL MIF”, lines 90-91, “…twice the concentration of 60 ng/mL used in the previous in vitro study [14];….”, and in line 300, “…study we administered 120 ng/ml MIF via the ventricle…”, it is not known how much of MIF were administered each mouse, e.g. mg of MIF in each Kg mouse.
3. Figure 3, revise sub part labelling from lower case alphabets to upper case alphabets.
4. “Scale, e.g. (10 µm)” is needed for Figure 3. Infarction Volume Changes (A) Representative images showed the actual size of the infarct through T2.
5. In line 214, “…..smaller than that in the MCAO+veh group (n = 13). * p < .000.”, “(*p < .000)” should have a value.
6. In line 259, “compared to the MCAO+veh (n = 12) (*p < .000). “, “(*p < .000)” should have a value.
7. The western blot results on IL-1β, IL-6 and TNF- α expression as shown in Figure 6, are not quite clear, the authors may need to improve the quality.
Author Response
Thank you for your great suggestion, please see the attachment for all the answers.

Round 2
Reviewer 1 Report
The authors have adequately addressed the raised comments. Thanks!